# Quantitative Identification of Internal and External Wire Rope Damage Based on VMD-AWT Noise Reduction and PSO-SVM

**DOI:** 10.3390/e24070981

**Published:** 2022-07-15

**Authors:** Jie Tian, Pengbo Li, Wei Wang, Jianwu Ma, Ganggang Sun, Hongyao Wang

**Affiliations:** 1School of Mechanical, Electronic and Information Engineering, China University of Mining and Technology (Beijing), Beijing 100083, China; tianj@cumtb.edu.cn (J.T.); 1786266175@163.com (W.W.); mjwzzgzhhsh@163.com (J.M.); sungg2020@163.com (G.S.); 2Key Laboratory of Intelligent Mining and Robotics, Ministry of Emergency Management, Beijing 100083, China

**Keywords:** internal wire rope damage, PSO-SVM algorithm, VMD-AWT noise reduction algorithm, wavelet energy entropy, non-destructive wire rope testing device

## Abstract

As a common load-bearing component, mining wire rope produces different types of damage during a long period of operation, especially in the case of damage inside the wire rope, which cannot be identified by the naked eye, and it is difficult to accurately detect such damage using the present technology. In this study we designed a non-destructive testing device based on leakage magnetism, which can effectively detect the internal defects of wire rope damage, and carried out simulation analysis to lay a theoretical foundation for the subsequent experiments. To address the noise reduction problem in the design process, a variational mode decomposition–adaptive wavelet thresholding noise reduction method is proposed, which can improve the signal-to-noise ratio and also calculate the wavelet energy entropy in the reconstructed signal to construct multi-dimensional feature vectors. For the quantitative identification of system damage, a particle swarm optimization–*support vector machine* algorithm is proposed. Moreover, based on the signal following the noise reduction step, seven different feature vectors, namely, the waveform area, peak value, peak-valley value, wavelet energy entropy classification, and identification of internal and external damage defects, have been determined. The results show that the device can be used to effectively identify internal damage defects. In addition, the comparative analysis showed that the algorithm can reduce the system noise and effectively identify internal and external damage defects with a certain superiority.

## 1. Introduction

Wire rope is widely used in mining operations due to its high strength, light weight, and good elasticity [1,2]. However, the degree of damage sustained by the wire rope increases considerably with the increase in the usage time and due to the increase in the long-term impact of factors such as tensile bending, alternating loads, and the environment. Furthermore, this damage is inevitable if it is not addressed in time, and it can adversely affect the productivity of mining operations and threaten the safety of both the personnel and the equipment. Coal mine safety regulations have been established to ensure the productivity of mining operations; according to these regulations, mining hoist ropes must be tested every day and their scrap period is two years. If the degree of damage does not exceed the relevant provisions, their usage can be extended by no more than one year.

Various methods have been proposed for the non-destructive testing of wire ropes. Most of the current studies are focused on methods such as ultrasonic detection [3], electromagnetic detection [4], X-ray detection, and magnetostriction [5], as well as eddy current, current, and vibration detection [6,7]. The electromagnetic detection method is the most widely implemented method, owing to its demonstrated reliability and practicality. The basic principle of the electromagnetic-based leakage detection method used in this study is shown in Figure 1. The permanent magnet magnetizes the wire rope to saturation, forming a closed magnetic circuit among the wire rope, magnet, and yoke. In the presence of a damage, the original magnetic induction lines through the wire rope form a closed magnetic circuit in the air and generate a leakage magnetic field.

When using the electromagnetic detection method to detect leakage, the wire rope detection signal is mixed with a variety of sources of interference noise, including the spiral structure of the wire rope, which produces periodic changes in the strand noise; the detection of the magnetic field in an environment of complex and variable high-frequency low-amplitude noise; the shaking of the wire rope during the operation process, producing low -frequency random noise; electromagnetic interference issuing from the electromagnetic detection circuit; detection line voltage jitter; drift; and other sources of noise, all of which affect the accurate judgment of the leakage signal. To address the aforementioned challenges, Peng, F. et al. [8] applied a multi-stage filtering method based on EEMD and optimal wavelets in three-dimensional UME signal processing to effectively suppress noise interference. Zhang, J. et al. [9] proposed a new filtering system consisting of the Hilbert yellow transform and compressive-aware wavelet filtering to denoise strand and high-frequency noises. Furthermore, Chun et al. [10] designed a filter based on the multi-stage wavelet analysis of a time-domain-reflection method. Moreover, they effectively eliminated the wild-point noise and industrial frequency interference noise. The abovementioned wire rope damage signal has been studied extensively. However, because the effect of wavelet packet decomposition depends on the choice of the wavelet basis function and the number of decomposition layers, it is not an adaptive signal decomposition method. In recent years, EMD has been widely used in mechanical fault diagnosis. However, owing to the existence of endpoint effects and modal confusion, this algorithm needs to be further studied. To address the limitations of EMD and WT, Dragomiretskiy et al. [11] proposed a new adaptive time-frequency analysis method called VMD in 2014. Compared with EMD and AWT, VMD can suppress interference signals, prevent the loss of useful information, and provide a high-quality data source for subsequent feature extraction. Moreover, it has high decomposition accuracy and operational efficiency and can effectively suppress the overlap mode in a signal decomposition process.

Wire rope detection is challenging because of signal noise reduction, as well as the difficulties involved in achieving a quantitative detection process following noise reduction, owing to the complex structure, shape, and location of the wire rope, which itself produces different types of defects. To solve this problem, some scholars have conducted representative studies. Li, J. et al. designed a nondestructive wire rope inspection device which used double detection plates to collect MFL data, improved the image resolution based on a super-resolution algorithm, and finally used the AdaBoost classifier to classify the defect images [12]. Zhang, J. designed a device based on a residual magnetic field device, proposing a novel filtering system to improve the signal-to-noise ratio, and at the same time used digital image processing techniques to achieve the quantitative recognition of defect images [13,14]. Tan, X. proposed a novel test structure with a huge array of magnetoresistive sensors to effectively identify multiple types of damage and finally applied radial basis neural networks for the quantitative recognition of magnetic images [15]. W Sharatchandra Singh et al. designed an ultrasonic sensor to detect wire rope damage signals by means of ultrasonic detection method and conducted quantitative recognition research using a BP neural network [16]. Artificial neural networks and related algorithms have contributed significantly to the field of pattern recognition. However, their recognition performance is significantly influenced by several parameters and can easily fall into a local minima in the optimization process. However, SVMs have few adjustable parameters and stable operation [17]. Thus, with fewer training samples, higher diagnostic accuracy can be achieved. Therefore, in this study we used SVMs based on PSO for the identification of internal and external wire rope damages.

In summary, it is difficult to detect the internal damage of wire ropes using the existing flaw detection equipment. Therefore, we have designed a wire rope detection device based on leakage magnetism. The detection device is implemented using permanent magnets to magnetize the wire rope, axial, and radial magnetization sensors in order to obtain the wire rope defect information. At the same time, the mapping relationship between internal damage and external damage was analyzed using the finite element method to prepare for the experiment. The VMD-AWT noise reduction method is used to reduce the noise of the original signal and calculate the wavelet information entropy based on the reconstructed signal to construct a multidimensional feature vector. Finally, the PSO-SVM algorithm is proposed to effectively and quantitatively classify and identify the internal and external defects of the wire rope using a multi-dimensional feature vector dataset.

The rest of this paper is organized as follows. In Section 2, we introduce the principles of VMD–AWT, wavelet energy entropy, and other methods. In Section 3, we introduce the principles of quantitative models (SVM and PSO–SVM). In Section 4, we present the simulation analysis carried out prior to the experiments. In Section 5, we introduce the experimental setup and experimental design process. In Section 6, we present the noise reduction of the experimental data via VMD-AWT and verify the signal-to-noise ratio by comparing it with others. In Section 7, we perform feature extraction and PSO-SVM identification and compare the results with those of various methods to verify the superiority of the algorithms in this study. Finally, the conclusions are summarized.

## 2. Noise Reduction and Recognition Principles

### 2.1. VMD-AWT Adaptive Noise Reduction

The proposed method uses the VMD-AWT approach to pre-process the wire rope experimental sampling data and employs the denoising principle of the VMD-AWT algorithm to decompose the original detection signal into k endowment modal components with different center frequencies and limited bandwidths. This is performed to achieve the correlation analysis of the decomposed IMF with the original signal and to perform adaptive wavelet threshold noise reduction on each IMF based on the correlation analysis results. Finally, the IMF components with adaptive wavelet threshold noise reductions are reconstructed to obtain notable wire rope damage characteristics.

The variational modal decomposition can be primarily divided into the construction of the variational problem and its solution; the construction of the variational problem is expressed as follows:

A constraint model is developed to minimize the sum of the bandwidths of the components in the VMD algorithm. The variational solution problem is expressed as
(1)$$\left\{\begin{array}{l} \underset{\left\{u_{k}\right\}\left\{\omega_{k}\right\}}{\min}\left\{\sum_{k}\|\partial_{t}\left[\left(\sigma \left(t\right)+\frac{j}{\pi t}\right)*u_{k}\left(t\right)\right]e^{-j\omega_{k}t}{\|}_{2}^{2}\right\}\\ \;s.t.\;\;\sum_{k}u_{k}=f \end{array}\right.$$

This constrained problem must be converted to an unconstrained problem to obtain the optimal solution of Equation (1), and this can be achieved by extending the Lagrange function as follows:(2)$$L\left(\left\{u_{k}\right\},\left\{\omega_{k}\right\},\lambda \right)=\alpha \sum_{k}{\left\|\partial_{t}\left[\left(\sigma \left(t\right)+\frac{j}{\pi t}\right)*\;u_{k}\left(t\right)\right]e^{-j\omega_{k}t}\right\|}_{2}^{2}+{\left\|f\left(t\right)-\sum_{k}u_{k}\left(t\right)\right\|}_{2}^{2}\;+\langle \lambda \left(t\right),f\left(t\right)-\sum_{k}u_{k}\left(t\right)\rangle \;$$

Here, α  represents the quadratic term penalty factor; λ represents the Lagrange multiplier; the alternating direction multiplier algorithm is used to continuously update each IMF’s $u_{k}^{n+1}$, and $\;w_{k}^{n+1}$, and $\;\lambda^{n+a}$; and the “saddle point” of the Lagrange expression is calculated.

Subsequently, the solution of the quadratic optimization problem to be solved is obtained as
(3)$${\hat{u}}_{k}^{n+1}\left(\omega \right)=\frac{\hat{f}\left(\omega \right)-\sum_{i\neq k}{\hat{u}}_{i}\left(\omega \right)+\frac{\hat{\lambda }\left(\omega \right)}{2}}{1+2\alpha {\left(\omega -\omega_{k}\right)}^{2}}$$
(4)$$\omega_{k}^{n+1}=\frac{\int_{0}^{\infty }\omega {\left|{\hat{u}}_{k}\left(\omega \right)\right|}^{2}d\omega }{\int_{0}^{\infty }{\left|{\hat{u}}_{k}\left(\omega \right)\right|}^{2}d\omega }$$
(5)$$\lambda^{n+1}=\lambda^{n}+\tau \left(x-\sum_{k}u_{k}^{n+1}\right)$$

Here, *n* denotes the number of iterations.
(6)$$\sum_{k=1}^{k}\frac{{\left|u_{k}^{n+1}-u_{k}^{n}\right|}_{2}^{2}}{{\left|u_{k}^{n}\right|}_{2}^{2}}<\varepsilon$$

Here, ε denotes the discriminatory accuracy.

A soft threshold approach is adopted here to perform noise reduction on the modal components following decomposition via the variational modal decomposition algorithm. The specific process is as follows.

A discrete wavelet transform transforms the function *f*(*t*) under the wavelet base. Its transform expression is given as
(7)$$T_{f}\left(\alpha ,\tau \right)=\left[f\left(t\right),\psi_{\alpha ,\tau }\left(t\right)\right]=\frac{1}{\sqrt{\alpha }}\int_{-\infty }^{+\infty }f\left(t\right)\psi^{*}(\frac{t-\tau }{\alpha })dt\;$$
where $\psi \left(t\right)$ is the wavelet base, α is the expansion, and τ is the translation. $WT_{f}\left(\alpha ,\tau \right)$ represents the wavelet coefficients under the wavelet transform. The inverse transform can be expressed as
(8)$$\left(t\right)=\frac{1}{c_{\phi }}\int_{0}^{+\infty }\frac{da}{\alpha^{2}}\int_{-\infty }^{+\infty }WT_{f}\left(u,\tau \right)\frac{1}{\sqrt{\alpha }}\psi \left(\frac{t-\tau }{a}\right)dt$$

Soft thresholding involves setting the smaller wavelet coefficients to zero and shrinking the larger coefficients toward the zero transform as follows:(9)$${\hat{W}}_{j,k}=\left\{\begin{array}{ll} sgn\left(W_{j,k}\right)*\left(\left|W_{j,k}-Thr\right|\right) & \left|W_{j,k}\right|\geq Thr\\ 0 & \left|W_{j,k}\right|<Thr \end{array}\right.$$
where $W_{j,k}$ denotes the wavelet coefficient and Thr denotes the wavelet threshold.

### 2.2. Wavelet Energy Entropy

The wavelet energy entropy consists of information entropy and the wavelet transform. The accuracy of the recognition results is further improved by the information entropy processing of the energy spectrum of the discrete wavelet transform subsignals.

Let $E_{jd}$ be the wavelet energy of the high-frequency component of the signal at scale j. It is expressed as
(10)$$E_{jd}=\sum_{t}{\left|x_{jd}\left(t\right)\right|}^{2}\;j=1,2,\cdots ,m.$$

The wavelet energy of the high-frequency component at each scale is summed. The energy set is given as
(11)$$E_{d}=\left\{E_{1d},E_{2d},\cdots ,E_{md}\right\}$$

To study the transformation law of the signal think $x\left(t\right)$ with time, a sliding window of width ω and a sliding factor δ is added to the jth subsignal; at this time, the intercepted signal is $x_{jd}\left(l\delta \right),\;x_{jd}\left(\omega +l\delta \right),\;l=1,2,\cdots ,L$, where L is the number of sliding windows.

In a certain sliding window, the total signal energy E is equal to the sum of the component energies $E_{jd}$ at each scale. Let $P_{j}=E_{jd}/E$; then, the wavelet energy entropy in the lth sliding window is
(12)$$H_{WEE,l}\left(E_{d}\right)=-\sum_{j}p_{j}lnp_{j}$$

Equation (12) reflects the characteristics of the signal energy distribution within a sliding window of the wavelet coefficients. Using the sliding window, the wavelet energy entropy variation pattern with time can be obtained.

## 3. Quantitative Analysis Model Construction

Artificial intelligence models are the most widely used models to establish the correlation between the internal damage, external damage, and feature quantities of wire ropes, using a large number of samples for learning and training. The increased use of support vector machines can be attributed to the structural risk minimization design. Furthermore, they significantly improve the classification performance, prevent over-learning, and present good generalization capability. Therefore, the support vector machine classification model was implemented for the identification of internal and external damage signals of wire ropes.

### 3.1. Support Vector Machine Principle

Support vector machines are commonly employed for classification and regression prediction. They are characterized by their ability to map the sample space to a higher-dimensional feature space through non-linear mapping. This allows them to determine the best separated hyperplane, to ensure that the sum of the distances from the two types of samples to the separated hyperplane are maximized, and to yield a good classification performance.

The algorithm steps are as follows:
(1)For a given training set, $T=\left\{\left(X_{1},Y_{1}\right),\left(X_{1},Y_{2}\right),\cdots ,\left(X_{n},Y_{n}\right)\right\}$ the computation is performed to obtain the best separated hyperplane, and the original classification problem is equated to an optimization problem that is to be solved, which can be expressed as:(13)$$\left\{\begin{array}{c} minJ\left(\omega ,b,\xi_{i}\right)=\frac{1}{2}\omega^{T}\omega +C\sum_{i=1}^{n}\xi_{i}\\ s.t.Y_{i}\left[\omega^{T}∲\left(X_{i}\right)-b\right]+\xi_{i}≽1\\ \xi_{i}≽0,i=1,2,\cdots ,n \end{array}\right.$$
where $X_{i}$ denotes the input vector, $Y_{i}$ denotes the corresponding label data, ω denotes the vector of separated hyperplane weights, b denotes the amount of deviation, and $\xi_{i}$ denotes the non-negative relaxation factor. C represents the penalty factor and $∲\left(X_{i}\right)$ represents a non-linear mapping function which maps the sample data to a high-dimensional feature space to avoid the problem of the linear indistinguishability of the data.(2)The Lagrangian functions constructed for the aforementioned equations based on the KKT conditions for ω and b are processed by taking partial derivatives and setting them to zero. Consequently, the optimal discriminant function can be solved, which is denoted as:(14)$$Y=\mathrm{sgn}\left\{\sum_{i=1}^{n}\propto_{i}Y_{i}\left[∲{\left(X\right)}^{T}∲\left(X_{i}\right)+b\right]\right\}$$
where $\propto_{i}\left(i=1,2,\cdots ,n\right)$ represents the Lagrangian multiplier.(3)The kernel functions for SVM $K\left(X_{i},\;X_{j}\right)=∲{\left(X_{i}\right)}^{T}∲\left(X_{j}\right)$ are then defined. The commonly used kernel functions for SVM include linear kernel functions, polynomial functions, radial basis functions (RBF), and sigmoid kernel functions. The RBF is selected as the kernel function of SVM since the RBF kernel contains fewer parameters, has relatively less of an impact on the complexity of the prediction model, and has a wider convergence domain and stronger generalization ability; it is expressed as follows:(15)$$K\left(X_{i},\;X_{j}\right)=\exp\left(-\gamma \|X_{i}-X_{j}{\|}^{2}\right)$$
where γ denotes the kernel function parameter.

### 3.2. PSO-Based Optimization of SVM Parameters

For the SVM classification model, which incorporates RBF as the kernel function, the error penalty factor, C, and the kernel function parameters, γ, are significant parameters which directly affect the recognition accuracy. The values of C and γ and the kernel function parameters must be optimized to improve the recognition rate of internal and external damages in wire ropes.

PSO displays fast convergence, a simple search mechanism, and good robustness in dynamic objective identification, which can prevent it from falling into local optimal solutions. Therefore, PSO was selected to update the parameters.

In the PSO algorithm, m particles are randomly generated to form the initial population, and each particle represents a feasible solution to the problem, where n is the number of dimensions in the solution space. The corresponding fitness $fit_{i}$ of each particle is the reciprocal of the sum of the squares of the errors, calculated based on an SVM multimetric mixture model. This can be expressed as
(16)$$fit_{i}=\frac{1}{\sum_{i=1}^{n}{\left(Y_{i}-{\hat{Y}}_{i}\right)}^{2}}$$
where $Y_{i}$ is the damage test value and ${\hat{Y}}_{i}$ is the value calculated using the SVM model.

The particle updates its position and velocity via individual and population extremes and determines the global optimal solution by following the current searched optimal value, which is expressed as follows:(17)$$v_{s,t+1}=\eta v_{s,t}+c_{1}r_{1}\left(u_{s,best}-u_{s,t}\right)+c_{2}r_{2}\left(u_{best}-u_{s,t}\right)$$
(18)$$u_{s,t+1}=u_{s,t}+v_{s,t+1}$$
where $\;u_{s}$ denotes the position of the *S*th particle in the search space, $v_{s}$ denotes the velocity of the *S*th particle, and *t* denotes the current number of updates. Furthermore, η denotes the inertia weight; $c_{1}$ and $c_{2}$ represent the acceleration factors; $r_{1}$ and $r_{2}$ represent random numbers between 0 and 1; $u_{s,best}$ denotes the current optimal position searched for by the *S*th particle; and $u_{best}$ denotes the current global optimal position searched for.

The parameter optimization procedures of the PSO-SVM prediction model are as follows:
(1)Normalize the data required for PSO-SVM training and prediction(2)Set the parameter values in the PSO algorithm and SVM model.(3)Initialize the particle population, calculate the corresponding fitness value of the particle according to Equation (16), and update its speed and position according to Equation (17).(4)During the process of continuous iteration in the search space, if the algorithm termination condition is satisfied, the optimal parameter is output; otherwise, step (3) is repeated.(5)The optimal parameters C and γ are used to train the SVM and build the PSO-SVM model to obtain the recognition results.

In summary, the identification process of the internal and external damages of a wire rope is shown in Figure 2.

## 4. Simulation of Internal and External Damage Thresholds

The damage in a wire rope under all movement conditions is simulated using dynamic simulation. This simulation helps in the process of obtaining a large amount of continuous damage data, which can be further used to calculate the theoretical threshold value of the wire rope under different damage conditions and provide the theoretical basis for further experiments.

Maxwell magnetic field simulation software was used to simulate the magnetic field of the wire rope testing model. The diameter of the wire rope was set to 30 mm. The simulated wire rope was a six-strand simulation model. The middle of the six-stranded wire rope interior contained the damage defect. The magnetic vector of the left magnet of the model diverged in the outward direction, and the magnetic vector of the right magnet contracted in the inward direction. This ensured that the wire rope formed a closed loop with the left and right magnets. The permanent magnet was a NdFeB magnet with a maximum magnetic energy product of about 380 $\mathrm{kA}/m^{3}$; the gag bit was composed of pure industrial iron. We set the maximum cell length to 1 mm, the maximum operation step to 10, the permissible error rate to 1%, and the residual value for non-linearity to 0.0001. The wire rope moved as a whole along the horizontal direction from left to right, from y = −40 mm to y = 40 mm; the step size was set to 0.1 mm. Figure 3 illustrates the theoretical simulation model.

### Analysis of Simulation Results

Figure 4 presents the simulation results. The different widths, lengths, and lift-off values (the distance between the sensor and the wire rope), the value of the wire rope’s internal damage defects, the defect type, and the correlations between these factors have been illustrated in the figure based on the principle of single variables.

Figure 4 shows that as both the wire rope defect type and the sensor distance from the location of the wire rope were different, the corresponding magnetic force line distribution state was different. This causes a defect length and a defect depth, and the lift-off value of the formation of the magnetic field strength is different. Notably, the magnetic flux density value of the defect magnetic signal increases with an increasing defect length. However, it tends to be stable when the defect length increases to a certain range. The magnetic flux density value of the defect’s magnetic signal increases with the defect depth, and the trend is linear. The flux density of the defect signal decreases with an increase in the lift-off value.

We produced results for 20 external defects and extracted the peak and valley values of the internal and external defects for comparison to determine the mapping correlation between the internal defects and external defects. Figure 5 presents the comparison results. The average value of the peak and valley values of the internal defects for different lengths of the wire rope was 0.0748 T. The average value of the peak and valley values of the external defects for different lengths of the wire rope was 0.1024 T. The relative difference between the peak and valley of the internal and external defects of different depths of the wire rope was approximately 26.95%. The internal defects of different depths of the average value was 0.12 T. The average value of the peaks and valleys of the external defects of the different depths was 0.169 T. The relative difference between the peaks and valleys of the internal and external defects of different depths of the wire rope was approximately 29%. The average value of the peaks and valleys of the internal defects was 0.03702 T. The average value of the peaks and valleys of the external defects was 0.0506 T, and the relative difference between these parameters was approximately 26.8%.

Based on what has been discussed above, a mapping correlation could be obtained between the different damages through the comparison of the internal and external defects of the wire rope’s peak and valley values; the two defects could be identified theoretically to provide a theoretical basis for the experimental design validation.

## 5. Experimental Verification

### 5.1. Experimental Platform Design

To verify the effectiveness of the proposed algorithm, in this study we designed an online wire rope inspection platform for experimental verification. The system comprised a magnetic leakage nondestructive testing device, an inspection circuit, a data acquisition module, and an overall support for the experiment. Figure 6 illustrates the overall structure.

### 5.2. Flaw Detector Prototype Design

The overall prototype of the leakage magnetic flaw detector was developed based on the available theoretical information, as shown in Figure 7. A modified Hall sensor was used for damage signal collection, and the collected signal was extracted in the acquisition system after it had passed through the signal acquisition box. The radial sensor was placed flat and evenly around the wire rope to obtain the radial magnetic field; this can effectively suppress the background magnetic field. The axial sensor collected the axial magnetic field signal, which was used in combination with the radial sensor. This improved the detection of wire rope defects along multiple directions and increased the accuracy. Consequently, the sensor presented high accuracy and low energy consumption.

### 5.3. Experimental Design

The working speed of the wire rope in the experiments was set to 1 m/s and the room temperature was set to 25 °C. The sampling was performed using an encoder, the parameters of which were set to 2000 p/r (p-pulse, r-ring).

The wire rope was selected for the experiment to test the magnetic field response at different defect lengths, different defect depths, and different lift-off values. The defect lengths were set to 3, 5, 7, 9, 11, 13, and 15 mm, and the damage cross-section was 10 mm. The defect depths were based on broken wires of 5, 10, 15, 20, 25, 30, and 35 mm, and the defect length was 15 mm. The different lift-off values were 2, 4, 6, 8, 10, and 12 mm, and the defect depth and defect length were 10 mm and 15 mm, respectively.

External damage defects of the wire rope were introduced to verify the difference in signals between the internal defects and external defects of the wire rope. The background of the experimental design of the external defects of the wire rope was consistent with that of the experiments of the internal defects of the wire rope, wherein 14 groups of internal defects and external defects of wire rope were produced, as shown in Figure 8.

## 6. Data Noise Reduction Processing

The experimental process with the use of the signal acquisition system to evaluate the internal damage of the wire rope is shown in Figure 9, where (a) indicates the internal overall defect signal of the wire rope, (b) the interception of the damage of a 15 mm internal defect signal, and (c) the interception of the internal defect-free signal. The external damage of the wire rope is shown in Figure 10, where (a) indicates the external overall defect signal of the wire rope, (b) the interception of the damage of a 15 mm external defect signal, and (c) the interception of the external defect-free signal. These figures show that the wire rope external defect signal was significantly different from the internal damage defect signal. Similarly, Figure 9c and Figure 10c show that the levels of signal noise for wire rope internal and external damage were considerable, were not conducive to the extraction of feature vectors, and affected the quantitative analysis of recognition. Therefore, the VMD-AWT algorithm for noise reduction was used to ensure the recognition rate of the internal and external damages.

### 6.1. VMD-AWT Algorithm to Decompose the Signal for Noise Reduction

The VMD-AWT algorithm was used for signal noise reduction processing. The VMD algorithm decomposes the signal by pre-determining the number of modes to be decomposed, K. Following the calculation, K lies in the range of 6–11 and is determined by observing that the center frequency of each mode generates modal blending, following the VMD process. Table 1 presents the center frequencies of the modal components under different values of the number of modes, K, after the damage signal was decomposed by VMD.

Table 1 shows that the center frequencies were close together when K = 10 and that an over-decomposition occurred; thus, a value of 9 was chosen for K. For the sake of simplicity, the subsequent analysis in this paper is focused only on the wire rope external length damage signal in order to demonstrate the display effect of the method. Figure 11 illustrates the wire rope damage signal obtained via VMD decomposition of the nine IMF components and their corresponding spectra, as well as the 10th residual for the VMD decomposition of the noise signal waveform and its corresponding spectrum.

It can be observed from the IMF component maps obtained via VMD decomposition and their corresponding spectrograms that the VMD algorithm can decompose the signal into multiple modal components with different center frequencies of finite bandwidth.

The noise in the signal was mostly observed at higher frequencies. The principle of noise reduction in the VMD algorithm involves decomposing the signal into multiple finite frequency bandwidths using different frequency bandwidths, and directly rejecting the decomposed high-frequency components to achieve noise reduction. However, since the decomposed high-frequency IMF was mixed with valid signals, directly rejecting the high-frequency components mixed with valid signals would result in the loss of effective information. Therefore, correlation contrast analysis was employed, and adaptive wavelet threshold noise reduction was then adopted for the IMF components of different frequencies, based on the correlation contrast analysis results, which eliminated most of the noise components while retaining the effective signal information.

The time-domain waveform obtained following the wavelet noise reduction is shown in Figure 12a, depicting the intercepted signal at 7 mm damage. Figure 12b depicts the overall time-domain waveform. Figure 12b shows that the changes in the waveforms of the VMD-AWT noise reduction signal and original signal were consistent, and the noise signal was removed. The effective features of the original signal were well retained, and the noise reduction effect was significant.

### 6.2. Comparison of Noise Reduction Effect

The advantages of the VMD-AWT method proposed in this paper in processing complex signals were demonstrated through comparison and analysis with AWT and EMD. Figure 13 presents the time-domain waveforms obtained following noise reduction, where (a) shows the overall time domain waveform following noise reduction and (b) shows the local signal waveform in the screenshot. Figure 13b shows that the signal curve following noise reduction obtained via VMD-AWT was smoother than that of other algorithms, and the noise component was smaller.

The denoising effect of the VMD-AWT on the damage signals of the wire rope was quantitatively evaluated by comparing and analyzing the VMD-AWT adaptive noise reduction algorithm with the AWT and EMD algorithms. Three metrics, i.e., the signal-to-noise ratio (*SNR*), mean square error (*RMSE*), and correlation coefficient (R), were used to evaluate the denoising effect.


(1)Signal-to-noise ratio (*SNR*):(19)$$SNR=10\log\frac{POWER_{signal}}{POWER_{noise}}$$


*SNR* reflects the noise-removal ability of a noise removal method; the larger the *SNR* value, the better the noise removal effect.


(2)Mean square error (*RMSE*):(20)$$RMSE=\sqrt{\frac{1}{n}\sum_{i=1}^{n}{\left(y_{i}-x_{i}\right)}^{2}}$$


The *RMSE* indicates the difference between the amplitude of the denoised signal and the original signal. The smaller the *RMSE* value, the better the denoising effect. Here, $y_{i}$ denotes the original signal and $x_{i}$ denotes the denoised signal.


(3)Correlation coefficient, R:(21)$$R=\mathrm{Cov}\left(x_{i},y_{i}\right)/\sigma_{x}\sigma_{y}$$


The correlation coefficient reflects the correlation between the denoised signal and the original signal. The closer R is to 1, the higher the correlation between the denoised signal and the original signal. Here, $\mathrm{Cov}\left(x_{i},y_{i}\right)$ represents the covariance of $x_{i}$ and $y_{i}$, and $\sigma_{x}$ and $\sigma_{y}$ represent the standard deviations of $x_{i}$ and $y_{i}$, respectively.

Table 2 presents a summary of the noise reduction effects of different noise reduction methods. It can be observed that the *SNR* of the VMD-AWT noise reduction method was approximately 27.5950 dB, which was higher than that of the AWT and EMD algorithms by 4.3427 dB and 6.2421 dB, respectively, with obvious noise reduction effects. Additionally, the correlation coefficient of the VMD-AWT noise reduction method was approximately 0.9910, which was close to the original signal in terms of morphology and which was thus able to eliminate the noise components and retain the effective features of the signal.

## 7. Quantitative Identification of Damage within PSO-SVM

### 7.1. Damage Signal Eigenvalue Extraction

The extraction of eigenvalues from the signal following noise reduction and reconstruction is beneficial to improving the classification recognition rate. However, as the damage-signal features of the wire rope present significant variability, the original sample data set of the wire rope’s characteristic wire-break-signal attributes had to be normalized using the following formula:(22)$$Y_{IJ}=\frac{X_{IJ}-{\mathrm{Xavg}}_{j}}{S_{j}},\;I=1,2,\cdots ,N;J=1,2,\cdots ,p$$
where $X_{IJ}$ represents the original feature attribute of the broken wire signal; ${\mathrm{Xavg}}_{j}$ represents the mean of the Jth feature attribute;${\;S}_{j}$ denotes its standard deviation; N denotes the total number of samples; p denotes the total number of feature attributes; and $Y_{IJ}$ denotes the normalized data value of the Jth feature attribute of the Ith sample.

The theoretical and experimental analyses indicated that the primary factors affecting the wire rope damage signals were the peak value, peak-to-peak value, area above the waveform, area below the waveform, wire rope diameter, and wire diameter. The experimentally obtained defect signals were used as samples to extract the different eigenvalues of each damage signal. The eigenvalues were combined with the wavelet energy entropy calculated following noise reduction to form multidimensional features, of which 20 groups of signal eigenvalue samples were extracted, as shown in Table 3.

### 7.2. Comparison of Internal and External Damage Identification

#### 7.2.1. PSO-SVM Damage Identification Results

We selected 106 groups from the extracted eigenvalue samples as training samples and 25 groups as test samples. We used the RBF kernel as the training function. The PSO algorithm was used to optimize the parameters of the penalty factor and kernel function of the SVM, and Figure 14 presents the obtained fitness curve. Figure 15 presents the recognition results, which demonstrate that the optimal fitness curve leveled off after several iterations. The number of populations was N = 20 and the number of terminated evolutionary iterations was 200; the acceleration constants were $C_{1}$ = 1.5 and $C_{2}$= 1.7, the optimal parameters obtained were C = 3.5817 and g = 181.5086, and the internal and external damage recognition rate was 97.619%.

#### 7.2.2. Comparison of Multiple Algorithms in Internal and External Damage Recognition

To verify the superiority of the proposed algorithm for the identification of damages inside and outside the wire rope, the same experimental data set was used for the comparative analysis of the EMD-AWT-PSOSVM, EMD-SVM, AWT-SVM, EMD-PSOSVM, and AWT-PSOSVM algorithms. The results are shown in Table 4.

To verify the effectiveness of the noise reduction algorithm, algorithms such as EMD-PSO-SVM and AWT-PSO-SVM were introduced for a comparison, as shown in Table 4. Additionally, after the proposed VMD-AWT algorithm was used to perform noise reduction, the recognition accuracy of the algorithm was significantly better than that of the other two algorithms. This indicates that the proposed algorithm can significantly improve the recognition rate of the PSO-SVM algorithm.

To verify the effectiveness of the PSO-SVM recognition process, four algorithms, namely, EMD-SVM, AWT-SVM, EMD-PSO-SVM, and AWT-PSO–SVM, were introduced. According to the longitudinal comparison, the recognition accuracy of the SVM after PSO was higher than that of the empirically selected SVM. This shows the effectiveness of the PSO-SVM algorithm.

In summary, compared with those of the other methods, the average recognition rate of the proposed algorithm was higher, at 97.1047 %; this indicates that the proposed algorithm can effectively identify internal damage defects. Thus, the superiority of the proposed algorithm was validated.

## 8. Conclusions

In this study, we designed a non-destructive-testing device for wire ropes and improved upon the Hall element detection device, as well as the detection accuracy of wire rope defects. Furthermore, based on the use of the non-destructive wire rope testing device for model construction and simulation analysis, the internal and external damages of the wire rope in the peak and valley values presented significant differences. Then, we extracted the internal and external damages in the peak and valley values and observed a theoretical mapping relationship between the two to lay a foundation for effective experimentation. For the quantitative identification of the internal and external damage values of the wire rope, an identification method based on the VMD-AWT and PSO–SVM algorithms was proposed. The experimental and comparative analyses demonstrated the superiority of the proposed method. The contributions of this study are as follows:
(1)Noise components still existed after the VMD decomposition of the noise signal. The wavelet threshold method was introduced to further process the noise components. The signal components were reconstructed to obtain the denoised signal via the identification of the useful components. In terms of morphology, we can effectively deal with the damage signal in the presence of sudden changes, spikes, and other nonlinear, local characteristics, and apply smoothing to retain the effective characteristics of the signal and adequately characterize the original signal, thereby improving the recognition rate of the damage signal inside and outside the wire rope.(2)Based on the damage signal obtained following noise reduction using the particle swarm algorithm to optimize the penalty factor and kernel function parameters of the SVM, seven different feature vectors, namely, the waveform area, peak, peak-valley, and wavelet energy entropy, were extracted through experimental and theoretical analyses to identify the internal and external damages of the wire rope. Compared with those of the SVM and PSO-SVM algorithms, the proposed algorithm displayed a superior identification performance.(3)The VMD-AWT noise reduction algorithm was compared with the AWT and EMD algorithms. From the comparative analysis values, the *SNR* of the VMD-AWT noise reduction method proposed in this study reached 27.5950 dB, which was higher than those of the AWT and EMD algorithms, which were 4.3427 and 6.2421 dB, respectively. Moreover, the noise reduction effect was more significant.(4)The experimental results showed that the proposed method was feasible, and the recognition rate of the VMD-AWT-PSO-SVM algorithm reached 97.619%, which could effectively identify the internal and external damages. Meanwhile, the comparison with the EMD-SVM, AWT-SVM, EMD-PSO-SVM, and AWT-PSO-SVM algorithms verified that the performance of this method was superior to that of other algorithms.

## Figures and Tables

**Figure 1 entropy-24-00981-f001:**
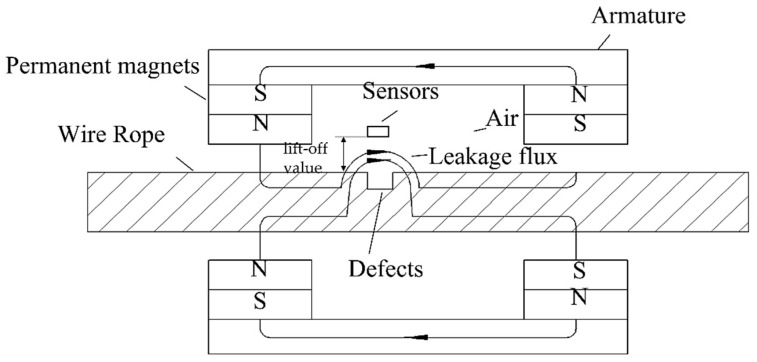
Basic principle of the magnetic leakage detection method.

**Figure 2 entropy-24-00981-f002:**
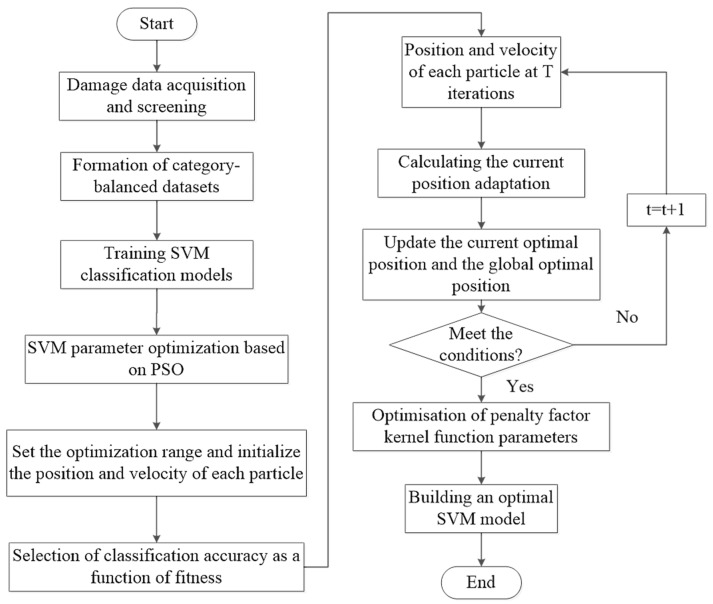
PSO-SVM classification process.

**Figure 3 entropy-24-00981-f003:**
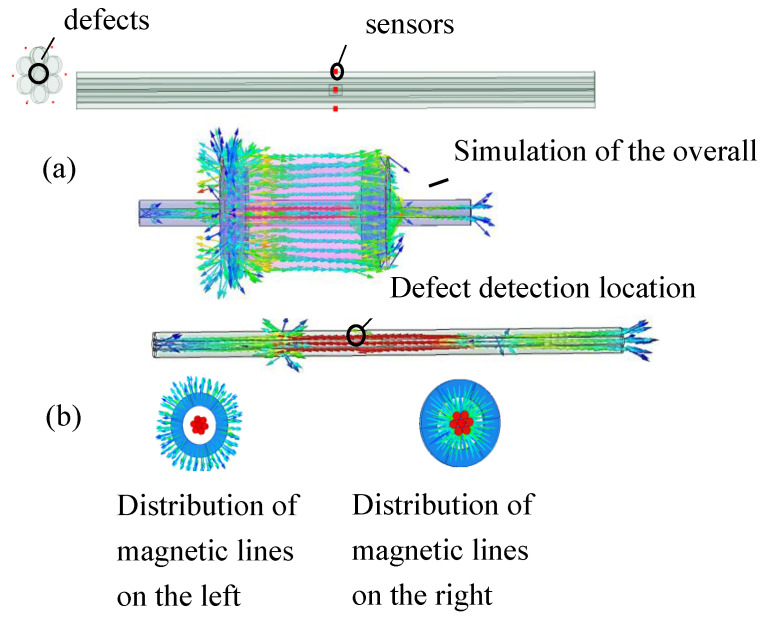
Theoretical imulation model. (**a**) Wire rope model, (**b**) simulated magnetic line distribution model.

**Figure 4 entropy-24-00981-f004:**
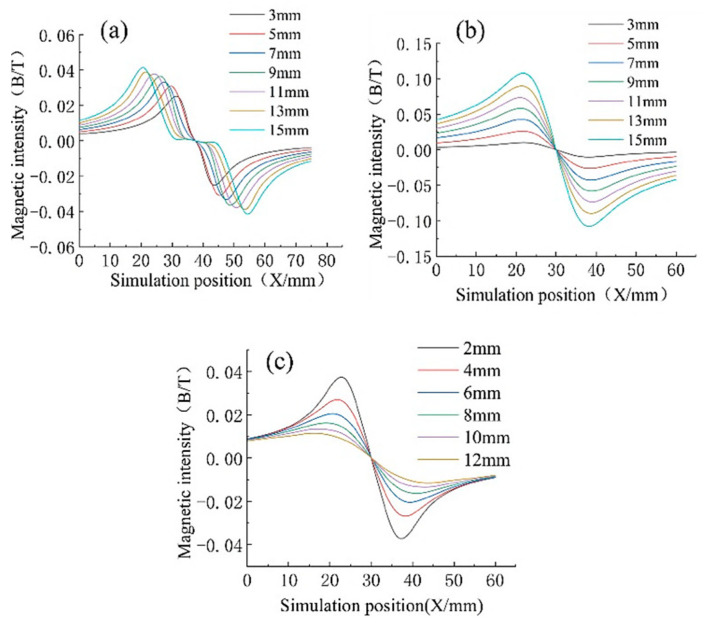
Simulation results of internal defects in wire ropes and (**a**) different widths, (**b**) different depths, and (**c**) different lift-off values.

**Figure 5 entropy-24-00981-f005:**
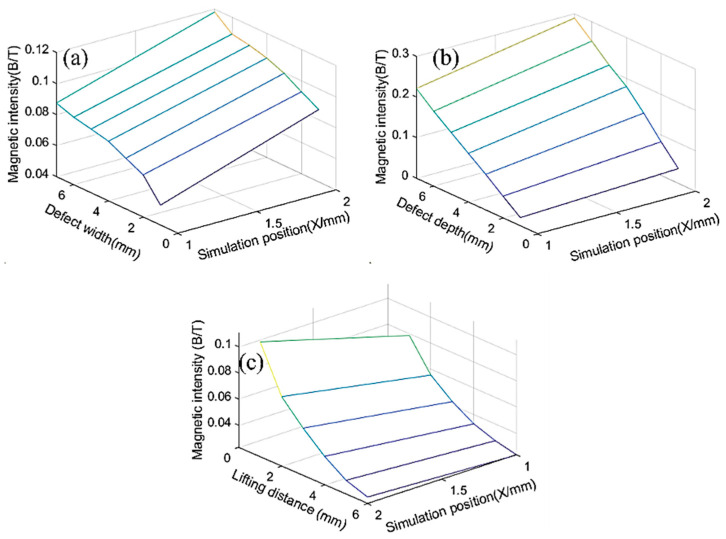
Comparison of peak and valley results. (**a**) Different lengths (**b**) Different depths (**c**) Different lift-off values.

**Figure 6 entropy-24-00981-f006:**
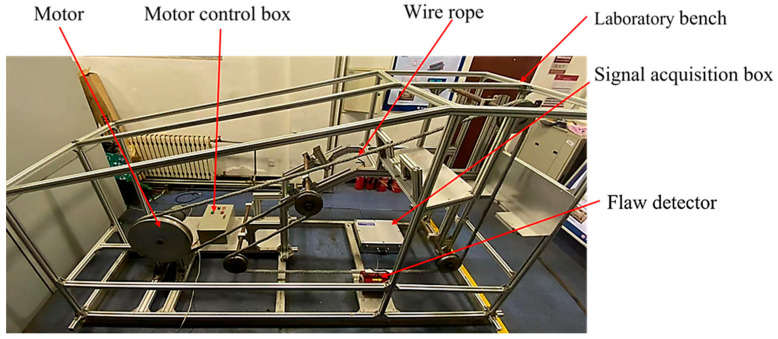
Online testing platform.

**Figure 7 entropy-24-00981-f007:**
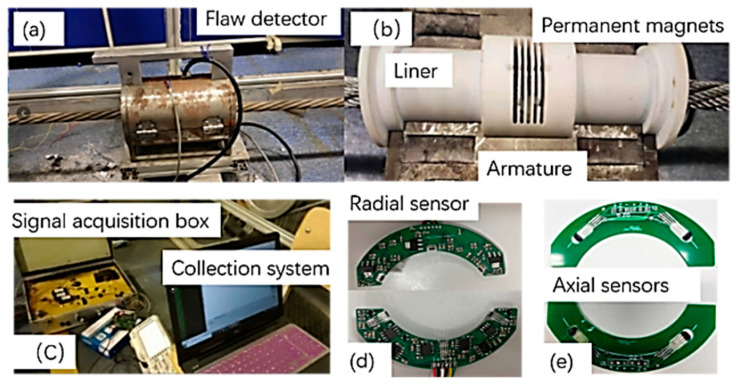
Wire rope inspection system. (**a**) Flaw detector, (**b**) internal structure of the detector, (**c**) overall system of signal acquisition, (**d**) radial sensor, (**e**) axial sensor.

**Figure 8 entropy-24-00981-f008:**
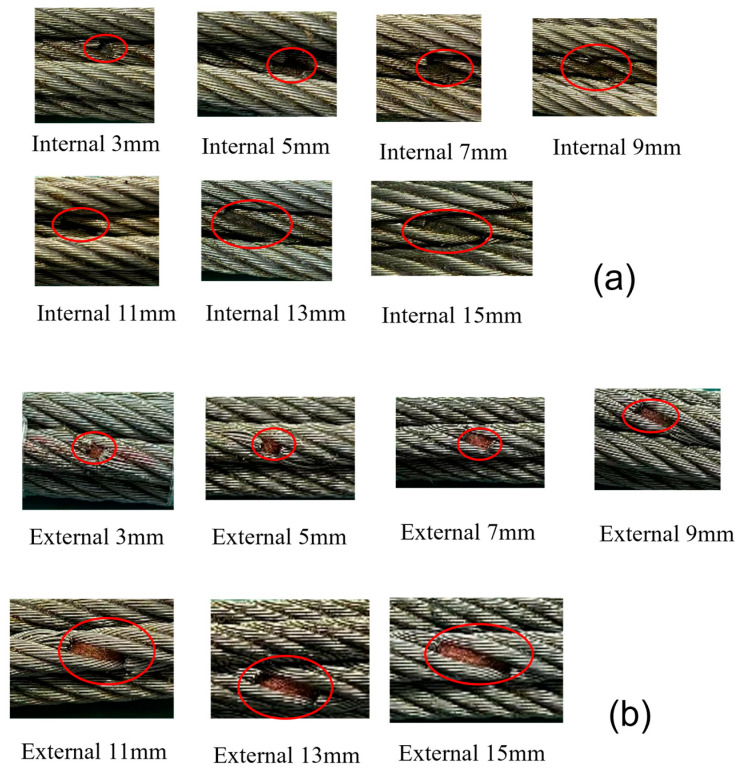
Experimental defects in steel wire rope. (**a**) Internal defect, (**b**) external defect.

**Figure 9 entropy-24-00981-f009:**
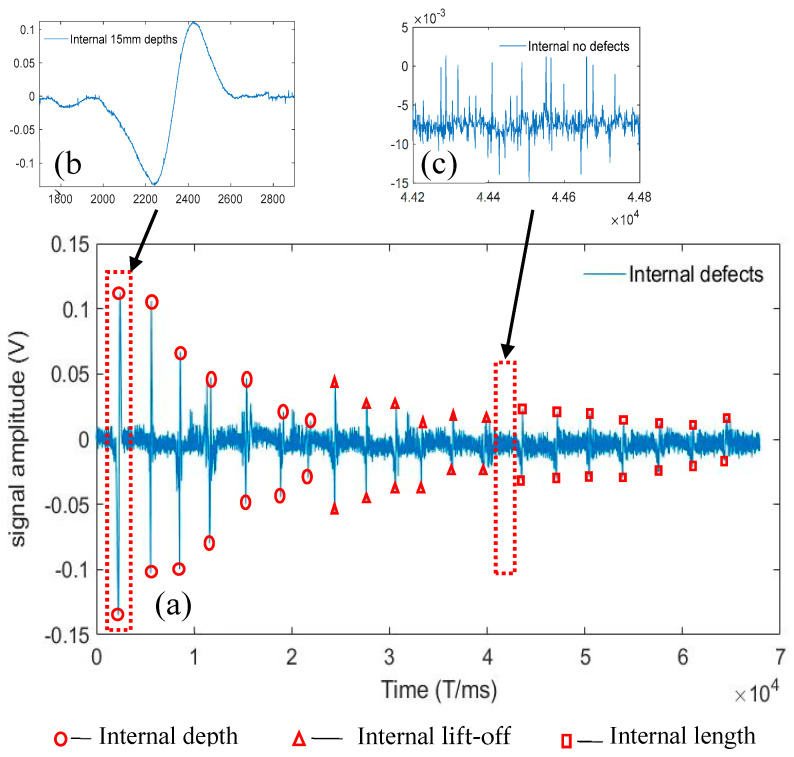
Internal defect of a wire rope: (**a**) the internal overall defect signal of the wire rope, (**b**) the interception of the damage from a 15 mm internal defect signal, and (**c**) the interception of an internal defect-free signal.

**Figure 10 entropy-24-00981-f010:**
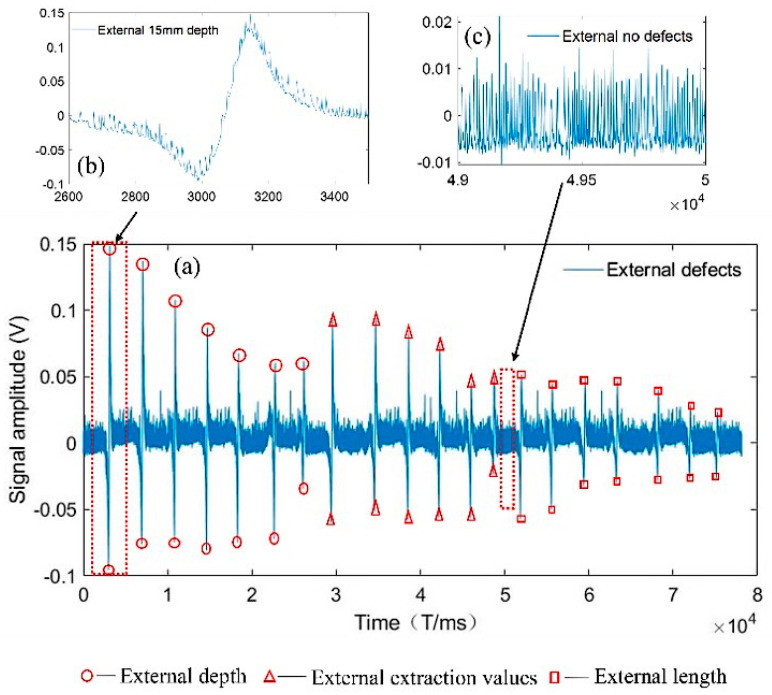
External defect of a wire rope: (**a**) the external overall defect signal of the wire rope, (**b**) the interception of the damage from a 15 mm external defect signal, and (**c**) the interception of an external defect-free signal.

**Figure 11 entropy-24-00981-f011:**
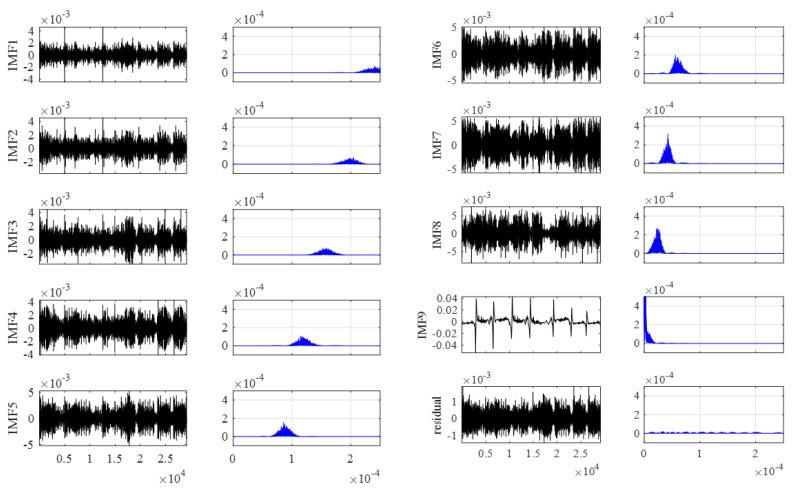
VMD decomposition diagram.

**Figure 12 entropy-24-00981-f012:**
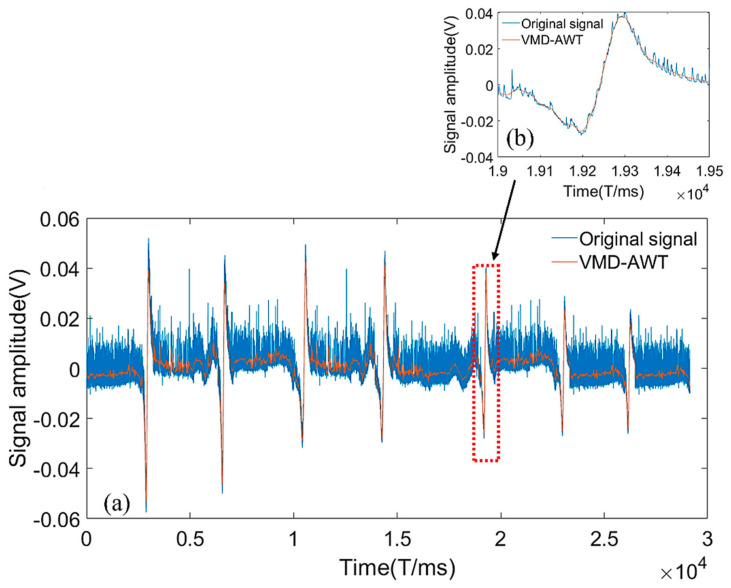
VMD-AWT time-domain noise reduction map. (**a**) Waveform plot after VMD-AWT noise reduction, (**b**) Comparison of VMD-AWT noise reduction and original local waveform.

**Figure 13 entropy-24-00981-f013:**
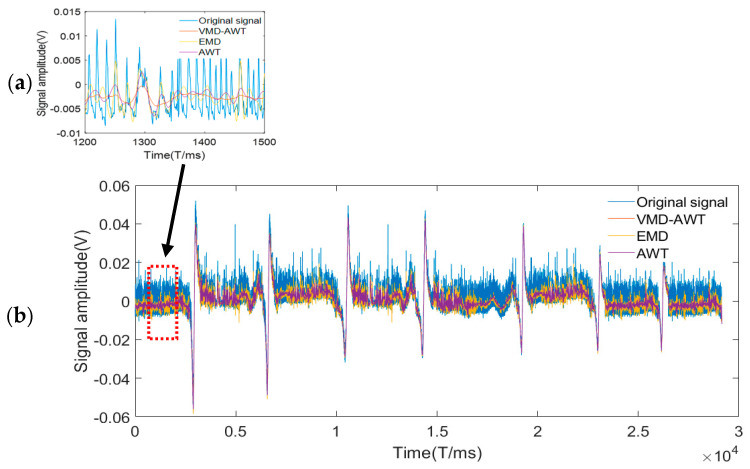
Multi-method comparison chart. (**a**) The overall time domain waveform after noise reduction, (**b**) the local signal waveform in the screenshot.

**Figure 14 entropy-24-00981-f014:**
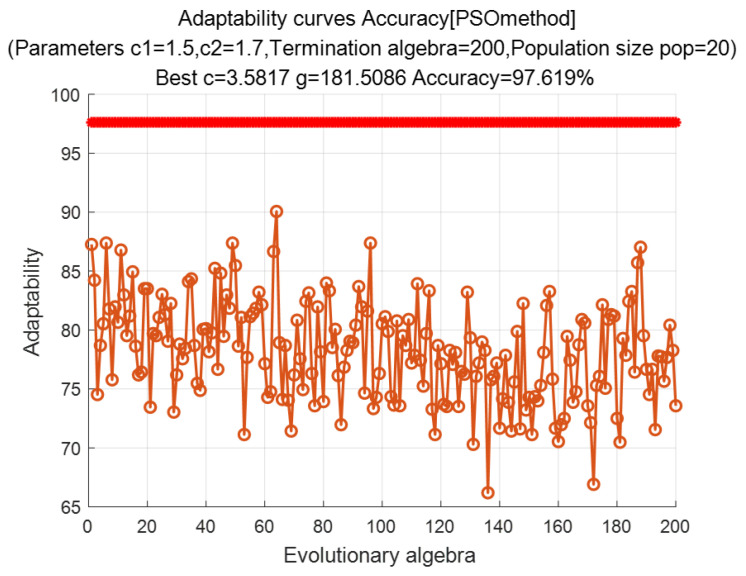
Adaptation curves for PSO optimization parameters.

**Figure 15 entropy-24-00981-f015:**
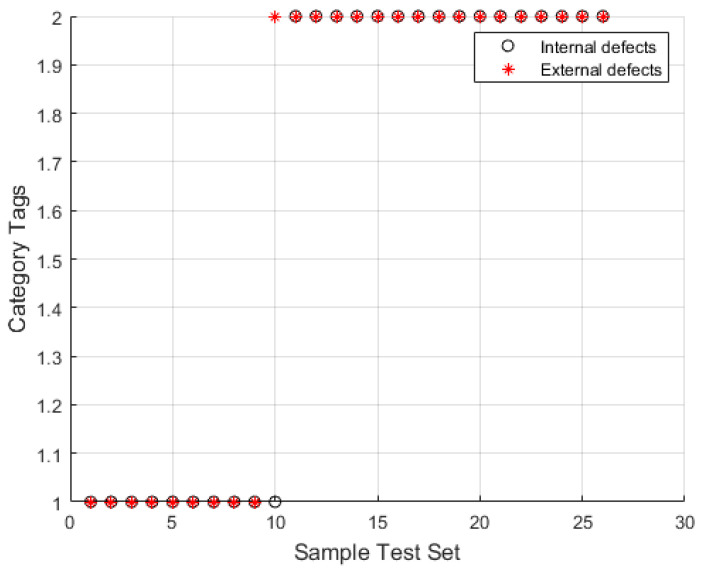
PSO-SVM algorithm classification results.

**Table 1 entropy-24-00981-t001:** Center frequencies corresponding to different K values.

K	Center Frequencies										
6	1.3 × 10^−10^	0.3527	0.668	0.9622	1.3366	1.7781					
7	1.3 × 10^−10^	0.3449	0.6537	0.9274	1.2252	1.5048	1.7858				
8	1.3 × 10^−10^	0.2200	0.4522	0.6944	0.9480	1.2311	1.5092	1.7862			
9	1.3 × 10^−10^	0.2148	0.4394	0.6807	0.9301	1.1952	1.4007	1.6140	1.7969		
10	1.3 × 10^−10^	0.2026	0.4092	0.6344	0.8138	0.9393	1.2153	1.4154	1.6212	1.7980	
11	1.3 × 10^−10^	0.1951	0.3145	0.5013	0.6694	0.8420	1.0222	1.2255	1.4226	1.6245	1.7984

**Table 2 entropy-24-00981-t002:** Noise reduction effects of different noise reduction methods.

Noise Reduction Indicators	*SNR*	*RMSE*	R
AWT	23.2523	0.0619	0.9903
EMD	21.3529	0.0656	0.9865
VMD-AWT	27.5950	0.0593	0.9910

**Table 3 entropy-24-00981-t003:** Signal eigenvalue samples.

Serial Number	Peak Value	Peak-to-Peak Value	Area below the Waveform	Area above the Waveform	Wavelet Energy Entropy	Wire Rope Diameter	Wire Diameter
1	0.04358	0.0939	12.504	−10.9718	0.0017	0.3	0.08
2	0.0457	0.08616	10.8723	−8.4009	0.0018	0.3	0.08
3	0.04447	0.08237	11.2564	−11.6126	0.0013	0.3	0.08
4	0.03313	0.06224	9.2805	−8.1743	0.0014	0.3	0.08
5	0.02583	0.05189	6.8754	−7.0832	0.0019	0.3	0.08
6	0.02418	0.04783	4.5371	−7.52	0.0015	0.3	0.08
7	0.1091	0.2407	24.2996	−24.7881	0.0024	0.3	0.08
8	0.09866	0.19352	13.5774	−8.7307	0.0042	0.3	0.08
9	0.0613	0.15481	5.4995	−15.9693	0.0056	0.3	0.08
10	0.04003	0.11305	13.9955	−15.9619	0.0082	0.3	0.08
11	0.03867	0.08336	24.379	−4.7879	1.62 × 10^−5^	0.3	0.08
12	0.0144	0.05574	5.6312	−4.8955	1.32 × 10^−5^	0.3	0.08
13	0.01078	0.0346	6.364	−3.4975	1.91 × 10^−5^	0.3	0.08
14	0.03816	0.09071	6.3293	−7.184	2.43 × 10^−5^	0.3	0.08
15	0.02118	0.06409	5.0068	−7.7042	1.29 × 10^−5^	0.3	0.08
16	0.01909	0.05717	1.6488	−11.6167	2.72 × 10^−5^	0.3	0.08
17	0.008588	0.046494	4.2232	−7.0306	2.42 × 10^−5^	0.3	0.08
18	0.01377	0.03742	6.8358	−3.3085	1.26 × 10^−5^	0.3	0.08
19	0.001713	0.024893	0.898	−9.769	1.94 × 10^−5^	0.3	0.08
20	0.01143	0.03476	1.9496	−5.2623	2.08 × 10^−5^	0.3	0.08

**Table 4 entropy-24-00981-t004:** Comparison results of multiple algorithms.

Serial Number	Algorithm	Identification Accuracy (%)					Average Identification Accuracy (%)
		1	2	3	4	5	
1	EMD-AWT-PSOSVM	97.6190	97.6190	97.6190	95.0476	97.6190	97.1047
2	EMD-SVM	61.3208	61.3208	58.4905	61.3208	61.3208	60.7547
3	AWT-SVM	87.7358	85.8491	87.7358	88.6792	87.7358	87.5471
4	EMD-PSOSVM	67.9245	67.9245	66.0377	67.9245	67.9245	67.5471
5	AWT-PSOSVM	92.4528	92.4528	90.0566	92.4528	92.4528	91.9735

## Data Availability

Not applicable.

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
