# Peer review of "Quantitative Identification of Internal and External Wire Rope Damage Based on VMD-AWT Noise Reduction and PSO-SVM"

_entropy, 2022, doi:10.3390/e24070981_

Round 1

Reviewer 1 Report

There are many abbreviations in the manuscript, the decoding of which is not given. For specialists in the field of signal processing, these abbreviations are generally accepted and widely used, but for a wide range of scientists, the lack of decoding of abbreviations makes it very difficult to understand the work.

So, in Chapter 3 “Quantitative Analysis Model Construction”, the concept of support vector machines appears (The increased use of support vector machines can be attributed to the structural risk minimization design.) Part 2.1 is entitled “3.1 Support Vector Machine Principle”, although the keywords include “PSO-SVM Algorithm”.  And earlier chapter 2 “Noise Reduction and Recognition Principles” contains the abbreviation "2.1 VMD-AWT Adaptive Noise Reduction".

I recommend the authors to provide a list of abbreviations in a separate appendix.

Author Response

Dear editors and reviewers:

Thank you for your letter and for the reviewers’ comments concerning our manuscript entitled “Quantitative identification of internal and external wire rope damage based on VMD–AWT noise reduction and PSO–SVM” (ID: entropy-1747722). Those comments are all valuable and very helpful for revising and improving our paper as well as providing x significant guidance to our research. We have carefully studied the comments and have made corrections, which we hope you will approve. The revised portions are highlighted in yellow in the paper. The main corrections in the paper and the responses to the reviewers’ comments are as follows:

Responser #1:

There are many abbreviations in the manuscript, the decoding of which is not given. For specialists in the field of signal processing, these abbreviations are generally accepted and widely used, but for a wide range of scientists, the lack of decoding of abbreviations makes it very difficult to understand the work.

So, in Chapter 3 “Quantitative Analysis Model Construction”, the concept of support vector machines appears (The increased use of support vector machines can be attributed to the structural risk minimization design.) Part 2.1 is entitled “3.1 Support Vector Machine Principle”, although the keywords include “PSO-SVM Algorithm”.  And earlier chapter 2 “Noise Reduction and Recognition Principles” contains the abbreviation "2.1 VMD-AWT Adaptive Noise Reduction".

Responds to the reviewer’s comments:

  1. Response to comment: (There are many abbreviations in the manuscript, the decoding of which is not given.)

Response: We have carefully completed the revision by adding an appendix at the end of the article, which is shown below.

Appendix

List of Abbreviations

VMD: Variational modal decomposition noise reduction algorithm

AWT: Wavelet noise reduction algorithm

PSO: Particle warm optimization algorithm

SVM: Support vector machine algorithm

EMD: Empirical modal decomposition noise reduction algorithm

VMD–AWT: Variational modal decomposition noise reduction algorithm based on wavelet adaptive filtering

PSO–SVM: Particle swarm optimization based on support vector machine

RBF: Radial basis function

MFL: Magnetic leakage detection

AdaBoost: Weak classifier algorithm

IMF: Eigen modulus obtained after variational modal decomposition

Hilber: Hilbert transform

Lagrange: Lagrangian function

SNR: Signal-to-noise ratio

RMSE: Mean square error

R: Correlation coefficient

EEMD: ensemble empirical mode decomposition

We are very sorry for our negligence of abbreviations decoding of which is not given. It's a really important question. We have made correction according to the Reviewer’s comments.

Special thanks to you for your good comments.

We have tried our best to make some changes and improve the manuscript. These changes will not influence the content and framework of the paper. Moreover, we did not list the changes but marked them in yellow in the revised paper.

We highly appreciate the editors’ and reviewers’ work and earnestly hope that the corrections will meet their approval.

Once again, thank you very much for your comments and suggestions.

Attached is the changed article.

Reviewer 2 Report

This paper presents an interesting methodology for detecting defects in wire ropes which are common structural elements for many industries. It is a challenging problem due in large part to the complicated mechanics of wire rope loading which makes simulation difficult. Measurements are similarly effected. Application of PSO optimized SVM to this problem should help to reduce these complications.

The authors should include a more detailed description of the literature with regard to PSO optimized SVM as well as provide some deeper explanation of these methods. There is decent descriptions (could be improved with maybe a simple graphic) of SVM, but there does not seem to be an adequate description of PSO that details how it is used to improve SVM classification. 

The conclusions do not synchronize well with the abstract or the introduction and should be put into a narrative form rather than a numbered list. The introduction could do a much better job of explaining the logic of the subsequent sections of the paper to give the reader an indication of how these sections relate to each other. In addition, the individual sections should provide segue into the next section to help the reader to understand the relationships between each section.

There are a significant number of formatting problems throughout the manuscript. Equations do not have consistent placement; tables seem to be missing headers and many of the figures showing important results are much to small to be read (even at high magnification which reduces resolution). I have attached a PDF that has notes and identifies the location of some of the formatting problems (but not all).

With regard to English language the early sections are stronger than some of the other sections. Some of the later sections would improve with attention by the strongest writer in the team. 

Author Response

Dear editors and reviewers:

Thank you for your letter and for the reviewers’ comments concerning our manuscript entitled “Quantitative identification of internal and external wire rope damage based on VMD–AWT noise reduction and PSO–SVM” (ID: entropy-1747722). Those comments are all valuable and very helpful for revising and improving our paper as well as providing x significant guidance to our research. We have carefully studied the comments and have made corrections, which we hope you will approve. The revised portions are highlighted in yellow in the paper. The main corrections in the paper and the responses to the reviewers’ comments are as follows:

Responser #2:

This paper presents an interesting methodology for detecting defects in wire ropes which are common structural elements for many industries. It is a challenging problem due in large part to the complicated mechanics of wire rope loading which makes simulation difficult. Measurements are similarly effected. Application of PSO optimized SVM to this problem should help to reduce these complications.

The authors should include a more detailed description of the literature with regard to PSO optimized SVM as well as provide some deeper explanation of these methods. There is decent descriptions (could be improved with maybe a simple graphic) of SVM, but there does not seem to be an adequate description of PSO that details how it is used to improve SVM classification. 

The conclusions do not synchronize well with the abstract or the introduction and should be put into a narrative form rather than a numbered list. The introduction could do a much better job of explaining the logic of the subsequent sections of the paper to give the reader an indication of how these sections relate to each other. In addition, the individual sections should provide segue into the next section to help the reader to understand the relationships between each section.

There are a significant number of formatting problems throughout the manuscript. Equations do not have consistent placement; tables seem to be missing headers and many of the figures showing important results are much to small to be read (even at high magnification which reduces resolution). I have attached a PDF that has notes and identifies the location of some of the formatting problems (but not all).

With regard to English language the early sections are stronger than some of the other sections. Some of the later sections would improve with attention by the strongest writer in the team. 

Responds to the reviewer’s comments:

1.Response to comment: (The authors should include a more detailed description of the literature with regard to PSO optimized SVM as well as provide some deeper explanation of these methods.)

Response: We have carefully revised the manuscript in on sideration of the raised questions, as follows.

3.2. PSO-Based Optimization of SVM Parameters

For the SVM classification model, which incorporates RBF as the kernel function, the error penalty factor, C, and the kernel function parameters,, are significant parameters which directly affect the recognition accuracy. The values of C and, and the kernel function parameters must be optimized to improve the recognition rate of internal and external damages in wire ropes.

PSO has a fast convergence, simple search mechanism, and good robustness in dynamic objective identification, which can prevent it from falling into local optimal solutions. Therefore, PSO is selected to update the parameters.

In the PSO algorithm,  particles are randomly generated to form the initial population, and each particle represents a feasible solution to the problem, where  is the number of dimensions in the solution space. The corresponding fitness  of each particle is the reciprocal of the sum of the squares of the errors calculated based on an SVM multimetric mixture model. It can be expressed as

,                                     (16)

where  is the damage test value and is the value calculated by the SVM model.

The particle updates its position and velocity via individual and population extremes and determines the global optimal solution by following the current searched optimal value, which is expressed as follows:

=+          (17)

+,                                        (18)

where denotes the position of the Sth particle in the search space,  denotes the velocity of the Sth particle, and t denotes the current number of updates. Furthermore,  denotes the inertia weight;  and  represent the acceleration factors;  and  represent random numbers between 0 and 1;  denotes the current optimal position searched for by the Sth particle; and  denotes the current global optimal position searched for.

 The parameters optimization procedures of the PSO–SVM prediction model are as follows:

  • Normalize the data required for PSO–SVM training and prediction.
  • Set the parameter values in the PSO algorithm and SVM model.
  • Initialize the particle population, calculate the corresponding fitness value of the particle according to Equation (16), and update its speed and position according to Equation (17).
  • During the process of continuous iteration in the search space, if the algorithm termination condition is satisfied, the optimal parameter is output; otherwise, step (3) is repeated.
  • The optimal parameters C and γ are used to train the SVM and build the PSO–SVM model to obtain the recognition results.

In summary, the identification process of the internal and external damages of the wire rope is shown in Figure 2.

Figure 2. PSO-SVM classification process.

  1. Response to comment: (The conclusions do not synchronize well with the abstract or the introduction and should be put into a narrative form rather than a numbered list.)

Response: We have carefully revised the conclusions and abstract as follows.

Abstract: As a common load-bearing component, mining wire rope will produce different types of damage during a long period of operation, especially the damage inside the wire rope cannot be identified by the naked eye, and it is difficult to be accurately detected by using the present technology. This paper designs a non-destructive testing device based on leakage magnetism, which can effectively detect the internal defects of wire rope damage, and at the same time carries out simulation analysis to lay a theoretical foundation for the subsequent experiments. To address the noise reduction problem in the design process, a variational mode decomposition–adaptive wavelet thresholding noise reduction method is proposed, which can improve the signal-to-noise ratio and also calculate the wavelet energy entropy in the reconstructed signal to construct multi-dimensional feature vectors. For the quantitative identification of system damage, a particle swarm optimization–support vector machine algorithm is proposed. Moreover, based on the signal following noise reduction, seven different feature vectors, namely the waveform area, peak value, peak-valley value, wavelet energy entropy classification, and identification of internal and external damage defects, are determined. The results show that the device can effectively identify internal damage defects. In addition, the comparative analysis shows that the algorithm can reduce the system noise and effectively identify internal and external damage defects with a certain superiority.

Conclusions

In this study, we designed a wire-rope non-destructive-testing device and improved a Hall element detection device, as well as the detection accuracy of wire rope defects. Further, based on the wire-rope non-destructive-testing device for model construction and simulation analysis, the internal and external damages of the wire rope in the peak and valley values presented a significant difference. Then, we extracted the internal and external damages in the peak and valley values and obtained a theoretical mapping relationship between the two to lay a foundation for effective experimentation. For the quantitative identification of the internal and external damage of the wire rope, an identification method based on the VMD-AWT and PSO–SVM algorithms was proposed. The experimental and comparative analyses showed the superiority of the proposed method. The contributions of this study are as follows:

(1) The VMD decomposition of the noise components still existed in the noise signal. The wavelet threshold method was introduced to further process the noise components. The signal components were reconstructed to obtain the denoised signal via useful components. In morphology, we can effectively deal with the damage signal in the presence of sudden changes, spikes, and other nonlinear, local characteristics, and apply smoothing to retain the effective characteristics of the signal and adequately characterize the original signal, thereby improving the recognition rate of the damage signal inside and outside the wire rope.

(2) Based on the damage signal following noise reduction using the particle swarm algorithm to optimize the penalty factor and kernel function parameters of the SVM,  s even different feature vectors, namely waveform area, peak, peak-valley, and wavelet energy entropy, are extracted through experimental and theoretical analyses to identify the internal and external damages of the wire rope. Compared with those of the SVM and PSO–SVM algorithms, the proposed algorithm has a superior identification performance.

(3) The VMD–AWT noise reduction algorithm was compared with the AWT and EMD algorithms. From the comparative analysis values, the SNR of the VMD– AWT noise reduction method proposed in this study reached 27.5950 dB, which was higher than those of the AWT and EMD algorithms, which were 4.3427 and 6.2421 dB, respectively. Moreover, the noise reduction effect was more significant.

(4) The experimental results showed that the proposed method was feasible, and the recognition rate of the VMD–AWT–PSO–SVM algorithm reached 97.619 %, which could effectively identify the internal and external damages. Meanwhile, the comparison with the EMD–SVM, AWT–SVM, EMD–PSO–SVM, and AWT–PSO–SVM verified that this method is superior to other algorithms.

Response : We have made careful changes to the formatting issues, as follows:

1.Line3 (Inconsistant formatting)

Response:Already modified.

  1. .Line4 (Inconsistant formatting)

Response:Already modified.

3.Line4 (provide reference)

Response:Appendices have been provided.

4.Line4 (provide reference)

Response:Appendices have been provided

5.Line4 (provide reference)

Response: Appendices have been provided.

  1. Line5(Inconsistent formatting)

Response: Already modified.

  1. Line5(Figures need to be scaled up)

Response: The picture has been updated.

  1. Line6(These images need to be a little larger. The axis labels and legend are a little hard to read)

Response: Already modified.

  1. Line6,the statements of“We produced” were corrected as“We produced results for”
  2. Line7,the statements of“This system is comprises” were corrected as“This system is comprised of...”
  3. Line7,“air”was deleted
  4. Line8(perhaps show fewer images in order to make the images larger at higher resolution.)

Response: The picture has been updated.

  1. Line8(These figures need a more detailed explanation. It would be helpful to show what the signal would look like in the absence of defects..)

Response: Modified to add two small images for comparison.

  1. Line8(This is I think a new section heading)

Response:Title has been added.

  1. Line8, the statements of “The VMD algorithm decomposes the signal by pre-determining the number of modes to be decomposed, K. Following the VMD decomposition, the centre frequencies of each mode produce modal blending to ultimately determine the number of modes to be decomposed, K.” were corrected as “The VMD-AWT algorithm is used for signal noise reduction processing, the VMD algorithm decomposes the signal by pre-determining the number of modes to be decomposed K. After calculation, the number of modes K is initially determined to be 6-11, and then the number of modes K to be decomposed is finally determined by observing that the center frequency of each mode generates modal blending after VMD decomposition.”
  2. Line9(Provide labels for columns and rows as is done for other tables)

Response: Tags have been added.

  1. Line9, the statements of“From Table 1 can be observed that the centre frequency of the 5th and 6th components is close to each other when k = 10. It can be assumed that there is an over decomposition; therefore, the number of modal, k, is set to 9, with the default value of 2000.” were corrected as “It can be seen from Table 1: the center frequencies are close together when k=10, and it can be considered that there is an over-decomposition, so the number of modes k is chosen as 9. for display effect.”
  2. Line10(Perhaps fewer images a larger scale and resolution. One of the figures seems to include tool bar icons from the capture tool that was used to produce the images)

Response: The picture has been updated.

  1. Line11(formatting problem)

Response: Already modified.

  1. Line11(inconsistent formatting)

Response: Already modified.

21.Line12, the statements of“line and data level of the wire rope wire break signal characteristic attributes pre-sents a relatively large difference in the data. Therefore, the original sample data set of the wire rope wire break signal characteristic attributes must be normalized using the following formula” were corrected as“Extraction of eigenvalues from the signal after noise reduction and reconstruction is beneficial to improve the classification recognition rate, but since the wire rope damage signal features present large variability, Therefore, the original sample data set of the wire rope wire break signal characteristic attributes must be normalized using the following formula.”

  1. Line12(inconsistent formatting)

Response: Already modified.

  1. Line13(Is this a new section or subsection?)

Response: Title has been added.

  1. Line14(is this a new section or new subsection? If so then the section numbering should be reflected in the formatting.)

Response: Title has been added.

  1. Line14(formatting problems)

Response: Already modified.

  1. Line14-15(This section should be significantly expanded and put into a more consistent narrative form rather than a numbered list. There are many interesting components of this paper but that the conclusion should highlight. The conclusion should be narratively tied to the introduction and abstract.)

Response: Conclusions to make the following updates:

In this study, we designed a wire-rope non-destructive-testing device and improved a Hall element detection device, as well as the detection accuracy of wire rope defects. Further, based on the wire-rope non-destructive-testing device for model construction and simulation analysis, the internal and external damages of the wire rope in the peak and valley values presented a significant difference. Then, we extracted the internal and external damages in the peak and valley values and obtained a theoretical mapping relationship between the two to lay a foundation for effective experimentation. For the quantitative identification of the internal and external damage of the wire rope, an identification method based on the VMD-AWT and PSO–SVM algorithms was proposed. The experimental and comparative analyses showed the superiority of the proposed method. The contributions of this study are as follows:

(1) The VMD decomposition of the noise components still existed in the noise signal. The wavelet threshold method was introduced to further process the noise components. The signal components were reconstructed to obtain the denoised signal via useful components. In morphology, we can effectively deal with the damage signal in the presence of sudden changes, spikes, and other nonlinear, local characteristics, and apply smoothing to retain the effective characteristics of the signal and adequately characterize the original signal, thereby improving the recognition rate of the damage signal inside and outside the wire rope.

(2) Based on the damage signal following noise reduction using the particle swarm algorithm to optimize the penalty factor and kernel function parameters of the SVM,  s even different feature vectors, namely waveform area, peak, peak-valley, and wavelet energy entropy, are extracted through experimental and theoretical analyses to identify the internal and external damages of the wire rope. Compared with those of the SVM and PSO–SVM algorithms, the proposed algorithm has a superior identification performance.

(3) The VMD–AWT noise reduction algorithm was compared with the AWT and EMD algorithms. From the comparative analysis values, the SNR of the VMD– AWT noise reduction method proposed in this study reached 27.5950 dB, which was higher than those of the AWT and EMD algorithms, which were 4.3427 and 6.2421 dB, respectively. Moreover, the noise reduction effect was more significant.

(4) The experimental results showed that the proposed method was feasible, and the recognition rate of the VMD–AWT–PSO–SVM algorithm reached 97.619 %, which could effectively identify the internal and external damages. Meanwhile, the comparison with the EMD–SVM, AWT–SVM, EMD–PSO–SVM, and AWT–PSO–SVM verified that this method is superior to other algorithms.

We have tried our best to make some changes and improve the manuscript. These changes will not influence the content and framework of the paper. Moreover, we did not list the changes but marked them in yellow in the revised paper.

We highly appreciate the editors’ and reviewers’ work and earnestly hope that the corrections will meet their approval.

Once again, thank you very much for your comments and suggestions.

Attached is the changed article.

Reviewer 3 Report

Dear authors,

the paper may be interesting, but the description of your methods and results is insuffient. Mathematically, a standart procedure is described, as far as  I understood, but the mathematics were not specified for your methods. The use of different wording, symbols and units makes it even more difficult to understand what you investigated. You did not explain the abbreviations of the title at all. The figures are too small and have a very poor quality. The signals do not show a critical noise/signal ratio. Why are your methods even necessary?

Best wishes

Author Response

Dear editors and reviewers:

Thank you for your letter and for the reviewers’ comments concerning our manuscript entitled “Quantitative identification of internal and external wire rope damage based on VMD–AWT noise reduction and PSO–SVM” (ID: entropy-1747722). Those comments are all valuable and very helpful for revising and improving our paper as well as providing x significant guidance to our research. We have carefully studied the comments and have made corrections, which we hope you will approve. The revised portions are highlighted in yellow in the paper. The main corrections in the paper and the responses to the reviewers’ comments are as follows:

Responser #3:

the paper may be interesting, but the description of your methods and results is insuffient. Mathematically, a standart procedure is described, as far as  I understood, but the mathematics were not specified for your methods. The use of different wording, symbols and units makes it even more difficult to understand what you investigated. You did not explain the abbreviations of the title at all. The figures are too small and have a very poor quality. The signals do not show a critical noise/signal ratio. Why are your methods even necessary?

Responds to the reviewer’s comments:

1.Response to comment: (Mathematically, a standart procedure is described, as far as I understood, but the mathematics were not specified for your methods.)

Response: We have carefully revised the proposed problem and all the methods are described mathematically, including VMD-AWT adaptive noise reduction, wavelet energy entropy, support vector machine and PSO-SVM algorithms are described mathematically.

  1. Response to comment: (You did not explain the abbreviations of the title at all)

Response: We have carefully completed the revision by adding an appendix at the end of the article, which is shown below.

Appendix

List of Abbreviations

VMD: Variational modal decomposition noise reduction algorithm

AWT: Wavelet noise reduction algorithm

PSO: Particle warm optimization algorithm

SVM: Support vector machine algorithm

EMD: Empirical modal decomposition noise reduction algorithm

VMD–AWT: Variational modal decomposition noise reduction algorithm based on wavelet adaptive filtering

PSO–SVM: Particle swarm optimization based on support vector machine

RBF: Radial basis function

MFL: Magnetic leakage detection

AdaBoost: Weak classifier algorithm

IMF: Eigen modulus obtained after variational modal decomposition

Hilber: Hilbert transform

Lagrange: Lagrangian function

SNR: Signal-to-noise ratio

RMSE: Mean square error

R: Correlation coefficient

EEMD: ensemble empirical mode decomposition

  1. Response to comment: (The figures are too small and have a very poor quality.)

Response: We have carefully finished revising and re-updating all the pictures, as there are too many pictures, please see the article for details.

  1. Response to comment: (he signals do not show a critical noise/signal ratio. Why are your methods even necessary?)

Response: We have carefully completed the modification and compared the VMD–AWT noise reduction algorithm with the AWT and EMD algorithms, and from the comparative analysis values, we can see that the SNR of the VMD–AWT noise reduction method proposed in this study reaches 27.5950 dB, which is higher than those of the AWT and EMD algorithms, which are 4.3427 and 6.2421 dB, respectively, and the noise reduction effect is significant, and the noise reduction effect of different noise reduction methods effects are shown in Table 1.

Table 1 Noise reduction effect of different noise reduction methods

Noise reduction index

SNR

RMSE

R

AWT

23.2523

0.0619

0.9910

EMD

21.3529

0.0656

0.9865

VMD-AWT

27.5950

0.0593

0.9903

To verify the superiority of the algorithm proposed in this study for the identification of damage inside and outside the wire rope, the same experimental data set was used for the comparative analysis of EMD–AWT–PSO–SVM, EMD–SVM, AWT–SVM, EMD–PSOSVM and AWT–PSO–SVM algorithms, and the comparative results are shown in Table 4.

Table 4 Comparison results of multiple algorithms

Serial number

Algorithm

Identification Accuracy (%)

Average Identification Accuracy (%)

1

2

3

4

5

1

EMD-AWT-PSOSVM

97.6190

97.6190

97.6190

95.0476

97.6190

97.1047

2

EMD-SVM

61.3208

61.3208

58.4905

61.3208

61.3208

60.7547

3

AWT-SVM

87.7358

85.8491

87.7358

88.6792

87.7358

87.5471

4

EMD-PSOSVM

67.9245

67.9245

66.0377

67.9245

67.9245

67.5471

5

AWT-PSOSVM

92.4528

92.4528

90.0566

92.4528

92.4528

91.9735

, In order to verify the effectiveness of the noise reduction algorithm, algorithms such as EMD–PSO–SVM and AWT–PSO–SVM are introduced for comparison, as shown in Table 4.  Moreover, after the proposed algorithm performs VMD–AWT noise reduction, the recognition accuracy of the algorithm is significant due to the other two algorithms, which verifies that the proposed algorithm can significantly improve the recognition rate of the PSO–SVM.

To verify the effectiveness of the PSO–SVM recognition, four algorithms, namely EMD–SVM, AWT–SVM, EMD–PSO–SVM, and AWT–PSO–SVM, are introduced. Moreover, following longitudinal comparison, the recognition accuracy of the support vector machine after particle swarm optimization is higher than that of the empirically selected support vector machine. This demonstrates the effectiveness of the PSO–SVM.

In summary, compared with those of other methods, the average recognition rate of the proposed algorithm in this study is 97.1047%, which is higher than the recognition rate of other methods. This indicates that the proposed algorithm can effectively identify internal damage defects. Thus, the superiority of the algorithm is validated.

  1. Response to comment: (There are a significant number of formatting problems throughout the manuscript.)

Response: We have made careful changes to the formatting issues, as follows:

1.Line1(Description of the developed methods and discussion of the results are missing. I do not understand what is new, what you did and why it is important. The wording and used signs as well as the units are incosequently used)

Response: the innovative work of this paper is:

(1) The VMD decomposition of the noise components still existed in the noise signal. The wavelet threshold method was introduced to further process the noise components. The signal components were reconstructed to obtain the denoised signal via useful components. In morphology, we can effectively deal with the damage signal in the presence of sudden changes, spikes, and other nonlinear, local characteristics, and apply smoothing to retain the effective characteristics of the signal and adequately characterize the original signal, thereby improving the recognition rate of the damage signal inside and outside the wire rope.

(2) Based on the damage signal following noise reduction using the particle swarm algorithm to optimize the penalty factor and kernel function parameters of the SVM, s even different feature vectors, namely waveform area, peak, peak-valley, and wavelet energy entropy, are extracted through experimental and theoretical analyses to identify the internal and external damages of the wire rope. Compared with those of the SVM and PSO–SVM algorithms, the proposed algorithm has a superior identification performance.

(3) The VMD–AWT noise reduction algorithm was compared with the AWT and EMD algorithms. From the comparative analysis values, the SNR of the VMD– AWT noise reduction method proposed in this study reached 27.5950 dB, which was higher than those of the AWT and EMD algorithms, which were 4.3427 and 6.2421 dB, respectively. Moreover, the noise reduction effect was more significant.

(4) The experimental results showed that the proposed method was feasible, and the recognition rate of the VMD–AWT–PSO–SVM algorithm reached 97.619 %, which could effectively identify the internal and external damages. Meanwhile, the comparison with the EMD–SVM, AWT–SVM, EMD–PSO–SVM, and AWT–PSO–SVM verified that this method is superior to other algorithms.

  1. Line1, the statements of “ However, the degree of damage sustained by the wire rope increases considerably with the increase in the usage time and due to the increase in the long-term impact of factors such as tensile bending, alternating load, and environment. Furthermore, this damage is inevitable if it is not addressed in time and can adversely affect the productivity of mining operations and threaten the safety of both the personnel and the equipment.” were corrected as “The VMD-AWT algorithm is used for signal noise reduction processing,the VMD algorithm decomposes the signal by pre-determining the number of modes to be decomposed K. Following the calculation, K lies in the range of 6–11 and is determined by observing that the center frequency of each mode generates modal blending following the VMD.”

3.Line2(Please explain AdaBoost)

Response: AdaBoost is a weak classifier algorithm and has been added to the appendix at the end of the article.

4.Line2 (what is leakage magnetism)

Response: The permanent magnet magnetizes the wire rope to saturation, forming a closed magnetic circuit among the wire rope, magnet, and yoke. In a damaged situation, the original magnetic induction lines through the wire rope form a closed magnetic circuit in the air and generate a leakage magnetic field.

This is reflected in the article introduction.

  1. Line 2(explain all abbreviations)

Response: Appendices have been provided

  1. Line3(equal? the mathematic expressions are incomplete)

Response: Already modified.

  1. Line3(this should be an integral, please use international standart for mathematic expressions, ∈ means belong to not epsilon ε, ≺,≻ Often used for denoting an order or, more generally, a preorder, when it would be confusing or not convenient to use < and >.Check the summary here: https://en.wikipedia.org/wiki/Glossary_of_mathematical_symbols)

Response: All the formulas have been re-edited.

  1. Line3(this is supposed to be an epsilon I guess. But the sign you wrote here means "belong to" ∈ )

Response: the statements of“?” were corrected as “”.

  1. Line5(figure does not suit the MDPI article design )

Response: Have reworked the image format, not sure if it's appropriate?

  1. (1)Line5(what were the parameters for this simulation?)

Response: The permanent magnet is NdFeB magnet, and the maximum magnetic energy product is about 380 , the armature is industrial pure iron .

  1. (2)Line5(What materials properties did you assume?)

Response:The maximum length of the unit is set to 1mm, the maximum operation step is 10, the allowed error rate is 1%, the residual value of non-linearity is set to 0.0001, the direction of movement of the wire rope is from left to right, and the wire rope moves along the horizontal direction from y = −40 mm to y = 40 mm; the step length is set to 0.1 mm.

  1. (3)Line5(how did you define the behaviour of the "leakage magnetic field" which is typically called: magentic flux leakage )

Response: The principle of magnetic leakage is mentioned in the introduction.

  1. Line5(why is "non-torsion" named explicitely here? Is torsion commonly simulated for rope wires? )

Response:"non-torsion" was deleted.

  1. Line5(What is an N48 permanent magnet?)

Response:the statements of“The N48 permanent magnet is used, which possesses a magnetic strength of 837,000 A/m;” were corrected as“The permanent magnet is a NdFeB magnet with a maximum magnetic energy product of approximately 380 “.

  1. Line5(megnetic strength is not a magnetic property. What‘s meant here? Remanence field strenght? Coercive field strength?)

Response: It has been changed to a maximum magnetic energy product.

  1. Line5(What is this information?!)

Response: “The model of the wire rope is 6 × 19S + FC” was deleted.

  1. Line5(What is this information?!)

Response: The image has been updated.

  1. Line6(What is the lift off value?)

Response: Distance between sensor and wire rope; already added in the text.

  1. Line6(discussion is missing. Why does the signal behave the way it does?)

Response:Figure 3 shows that as both the wire rope defect type and the sensor distance from the location of the wire rope are different, the corresponding magnetic force line distribution state is different. This causes a defect length and a defect depth. In addition, the lift-off value of the formation of the magnetic field strength is different. Notably, the magnetic flux density value of the defect magnetic signal increases with increasing defect length. However, it tends to be stable when the defect length increases to a certain range. The magnetic flux density value of the defect magnetic signal increases with the defect depth, and the trend is linear. The flux density of the defect signal decreases with an increase in the lift-off value.

  1. Line6(Now you use Tesla. The usage of magnetic properties historically is confusing. I recommend to use SI Units)

Response:We have not found the relevant conversion formula. Please tell us how to change it; can the units here not be modified?

  1. Line7(Figure too small. I can identify an aluminium profile, but nothing else)

Response: The image has been updated.

  1. Line7(what flaw?Do you mean flux?)

Response: the statements of “the flaw detector” were corrected as “magnetic leakage nondestructive testing device.”

  1. Line7(what is this unit?)

Response:p-pulse, r-ring. This is reflected in the article.

  1. Line7(mm?)

Response: the statements of“wires”were corrected as “mm.”

  1. Line7(I still don’t know what "lift-off" is)

Response: Distance between sensor and wire rope; this has been added to the text.

  1. Line8(too small)

Response: The picture has been updated.

  1. Line10(Poor Quality. Obviously screenshots, as the "tools" of your software are visible for IMF7.)

Response: The picture has been updated.

  1. Line10(too small. I don’t see a difference between the signals)

Response: The picture has been updated.

  1. Line10(too small.)

Response: The picture has been updated.

  1. Line12(too small.)

Response: The format has been modified.

  1. Line14(this method is not new. These kind of experiments were published decades ago.)

Response:The innovation point has been removed.

  1. Line15(Different signals depending on the position and type of a defect is not really suprising, rigtht? Additionally, a discussion is missing that explains the observed behaviour.)

Response:A discussion has been added to the article.

  1. Line15(You showed 2 methods for signal manipulation. There is no discussion about the quality compared to other methods. I still don’t know what VMD-AWT stands for.)

Response:The VMD-AWT has been described in detail in the principle and is also described in the appendix.

  1. Line15(Description of the developed method and discussion of the results are missing)

Response:Already added “the developed method and discussion of the results”.

We have tried our best to make some changes and improve the manuscript. These changes will not influence the content and framework of the paper. Moreover, we did not list the changes but marked them in yellow in the revised paper.

We highly appreciate the editors’ and reviewers’ work and earnestly hope that the corrections will meet their approval.

Once again, thank you very much for your comments and suggestions.

Attached is the changed article.

Round 2

Reviewer 3 Report

Dear Authors,

there are still spelling errors (e.g. PSO: Particle (s)warm optimization in the appendix). The quality of the images is low, but at least readable.

Please do a very careful spell check and further improve the images quaity, if possible.

Best wishes

Author Response

Dear editors and reviewers:

Thank you for your letter and for the reviewers’ comments concerning our manuscript entitled “Quantitative identification of internal and external wire rope damage based on VMD–AWT noise reduction and PSO–SVM” (ID: entropy-1747722). Those comments are all valuable and very helpful for revising and improving our paper as well as providing x significant guidance to our research. We have carefully studied the comments and have made corrections, which we hope you will approve. The revised portions are highlighted in yellow in the paper. The main corrections in the paper and the responses to the reviewers’ comments are as follows:

Responser #3:

there are still spelling errors (e.g. PSO: Particle (s)warm optimization in the appendix). The quality of the images is low, but at least readable.

Please do a very careful spell check and further improve the images quaity, if possible.

Responds to the reviewer’s comments:

  1. Line 9(The quality of the images is low, but at least readable.)

Response:It has been revised, especially Figure 6 using a clearer picture.

2.Line18(spelling errors (e.g. PSO: Particle (s)warm optimization in the appendix)

Response:The article has been checked and revised.

Special thanks to you for your good comments.

We have tried our best to make some changes and improve the manuscript. These changes will not influence the content and framework of the paper. Moreover, we did not list the changes but marked them in yellow in the revised paper.

We highly appreciate the editors’ and reviewers’ work and earnestly hope that the corrections will meet their approval.

Once again, thank you very much for your comments and suggestions.

Attached is the revised manuscript.
